# Possible Event-Related Potential Correlates of Voluntary Attention and Reflexive Attention in the Emei Music Frog

**DOI:** 10.3390/biology11060879

**Published:** 2022-06-08

**Authors:** Wenjun Niu, Di Shen, Ruolei Sun, Yanzhu Fan, Jing Yang, Baowei Zhang, Guangzhan Fang

**Affiliations:** 1Chengdu Institute of Biology, Chinese Academy of Sciences, Chengdu 610041, China; wenjunniu@163.com (W.N.); akshendi@163.com (D.S.); fanyz@cib.ac.cn (Y.F.); yieongjing@163.com (J.Y.); 2School of Life Science, Anhui University, Hefei 230601, China; sunruolei@hisense.com (R.S.); zhangbw@ahu.edu.cn (B.Z.); 3College of Life Sciences, University of Chinese Academy of Sciences, 19A Yuquan Road, Beijing 100049, China

**Keywords:** auditory perception, voluntary attention, reflexive attention, event-related potentials (ERP), music frog

## Abstract

**Simple Summary:**

We investigated auditory event-related potentials (ERP) related to auditory attention in music frogs. Our objective was to explore whether ERP components related to voluntary attention and reflexive attention exist in frogs. We found that the amplitudes of stimulus preceding negativity (SPN, related to voluntary attention and under up-down control) evoked by silence replacement in the telencephalon were the largest when the sequence of acoustic stimuli could be predicted, while the N1 amplitudes (related to reflexive attention and under bottom-up control) evoked in the mesencephalon were the largest when the sequence of acoustic stimuli could not be predicted. This suggests that human-like ERP components related to voluntary attention and reflexive attention exist in the lower vertebrates also.

**Abstract:**

Attention, referring to selective processing of task-related information, is central to cognition. It has been proposed that voluntary attention (driven by current goals or tasks and under top-down control) and reflexive attention (driven by stimulus salience and under bottom-up control) struggle to control the focus of attention with interaction in a push–pull fashion for everyday perception in higher vertebrates. However, how auditory attention engages in auditory perception in lower vertebrates remains unclear. In this study, each component of auditory event-related potentials (ERP) related to attention was measured for the telencephalon, diencephalon and mesencephalon in the Emei music frog (*Nidirana daunchina*), during the broadcasting of acoustic stimuli invoking voluntary attention (using binary playback paradigm with silence replacement) and reflexive attention (using equiprobably random playback paradigm), respectively. Results showed that (1) when the sequence of acoustic stimuli could be predicted, the amplitudes of stimulus preceding negativity (SPN) evoked by silence replacement in the forebrain were significantly greater than that in the mesencephalon, suggesting voluntary attention may engage in auditory perception in this species because of the correlation between the SPN component and top-down control such as expectation and/or prediction; (2) alternately, when the sequence of acoustic stimuli could not be predicted, the N1 amplitudes evoked in the mesencephalon were significantly greater than those in other brain areas, implying that reflexive attention may be involved in auditory signal processing because the N1 components relate to selective attention; and (3) both SPN and N1 components could be evoked by the predicted stimuli, suggesting auditory perception of the music frogs might invoke the two kind of attention resources simultaneously. The present results show that human-like ERP components related to voluntary attention and reflexive attention exist in the lower vertebrates also.

## 1. Introduction

Attention, being central to cognition [1], refers to selectively processing information relevant to the current task while ignoring other irrelevant information [2]. Depending on the direction of information flow, attention can be divided into two types: voluntary attention and reflexive attention [3,4]. The former under top-down control can direct attention to the position related to the goal [5,6], while the latter under bottom-up control is often referred to as exogenous attention because it is thought to be triggered by external stimuli [3,6]. Attention engages in different modalities including visual and auditory, and in various stages of brain functions from processing and perception of information to finally behavioral response [7]. Previous studies have found that voluntary attention and reflexive attention struggle to control the focus of attention with interaction in a push–pull fashion [2] in higher vertebrates including birds [8], mammals [9], non-human primates [10] and humans [11]. For example, behavioral evidence has shown that a sudden alarm call will cause Japanese great tits (*Parus minor*) to stop searching for food and quickly glance at their surroundings [12].

For many animal species (including humans), multiple cortical and subcortical structures engage in auditory attention, and neural mechanisms of attention select the information that can gain access to the brain networks making cognitive decisions [1]. Both forebrain and midbrain networks contain specialized neural circuits that process the highest priority information at each stage for decision-making. The former selects information, based on task demands, from all available sources, including sensory input, memory stores, and plans for action, and then assign attention either to stimulus features, sensory modalities, objects, locations, or memory stores [1]. Conversely, the latter is concerned only with the relative priorities of locations, based on the stimulus’ physical salience and its behavioral relevance, and assigns spatial attention to the highest priority location [1]. Moreover, overlapping brain networks, such as the fronto-parietal network, can be activated by bottom-up triggered and top-down controlled auditory attention to pitch [13,14,15], suggesting the information involved in the two types of attention would be integrated in the brain. However, it is not yet clear whether voluntary attention and reflexive attention exist in the lower vertebrates.

For most anuran species, vocal communication is the most important medium for reproductive success and social interactions [16,17]. Generally, various species always gather in choruses to attract conspecific mates. Therefore, the acoustic environment of a chorus would be very complex due to high levels of background noise, vagaries of spatial distribution of males, intense competition between males, and temporal overlap among advertisement calls of rivals. The noisy social environments created by a chorus may affect communication efficiency [18]. For example, overlapping calls might obscure fine acoustic attributes of calls and further influence signal selectivity and decision-making in females [19,20]. Accordingly, how to detect conspecific vocalizations in the choruses and respond to them correctly is a main challenge for receivers [17,21,22,23]. The solution adopted by males of some anuran species is to use selective attention to adjust note or call timing according to only the loudest (or nearest) one or two neighbors in the chorus while ignoring the notes or calls of other individuals [24,25]. Both call alternation and call synchrony may result from a neural process that may reset a male’s call-timing after perception of a rival’s call [26,27]. By this “inhibitory-resetting” mechanism of call-timing, a male can increase the likelihood of occupying a leading position relative to those of his neighbors, with which the male can compete effectively and attract the attention of females because of the precedence effect, an inherent property of the auditory system in vertebrates [28,29]. In fact, computer modeling has demonstrated that both the “inhibitory-resetting” mechanism and selective attention may be favored by selection when female mate choice is biased by the precedence effect [27]. In addition, a sudden sound such as loud voices from a person would cause the calling frogs to stop calling. Consequently, it is, therefore, reasonable to assume that both voluntary attention and reflexive attention might engage in auditory perception in anurans; however, there is still much that remains unknown about whether these two attention patterns could be reflected by brain activities in anurans.

Event-related potential (ERP) is the measured brain response to a specific event, whose amplitudes and latencies can be used to examine processing efficiency and time course of information processing in the brain [30]. In humans, the main components of auditory ERP include N1, P2 and P3, with their peaks at about 100, 200 and 300 ms after the stimulus onset, respectively [31,32,33], during which N1 relates to attention of the subject and is sensitive to physical features of the stimulus [33,34]. In addition, the stimulus preceding negativity (SPN) isolated from contingent negative variation is thought of as a cognitive component because its amplitude would gradually increase with the approach of the stimulus. Therefore, the SPN could be used as a measure of expectations with voluntary attention and a tool of the research on the impact of uncertainty [35,36,37]. Interestingly, human-like auditory ERP components, with different latencies across various species for each component, have been determined in monkeys [38], cats [39], dolphins [40], rabbits [41], rats [42] and frogs [43,44,45,46,47]. Because important neuroanatomical features, including a set of brain structures that attention depends on, have been conserved during vertebrate brain evolution [48,49,50], similar ERP components across different species may indicate similar brain functions to some extent.

The Emei music frog (*Nidirana daunchina*) is a typical seasonal reproductive species. Males of this species produce advertisement calls from inside and outside underground burrows in the breeding season [51]. Calls produced from inside burrows are of high sexual attractiveness for conspecific females because of the call acoustics modified by resonant properties of the burrows, while the calls produced from open fields are weakly sexually attractive. Previous behavioral and electrophysiological studies on this species have revealed that auditory perception in this species might recruit selective attention [43,51,52]. For example, females prefer inside calls to outside ones in phonotaxis tests, while males are more likely to compete vocally against inside calls compared to outside ones, congruent with the findings that voluntary attention may be involved in anurans’ auditory perception [24,25]. At the electrophysiological level, inside calls could evoke significantly greater N1 amplitudes compared with outside calls when the two type of calls are played back according to a random sequence [43], suggesting reflexive attention may exist in music frogs. Based on these studies, we hypothesized that auditory perception in this species would depend on the combination of voluntary attention and reflexive attention. To verify this, we used a binary playback paradigm with silence period replacement and an equiprobably random playback paradigm to explore these two types of attention, respectively. During broadcasting of acoustic stimuli related to breeding or survival, electroencephalogram (EEG) signals were synchronously collected from both sides of the telencephalon, diencephalon and mesencephalon, and the amplitude and latency were acquired for each ERP component. We predicted that: (1) if voluntary attention exists, the SPN amplitudes elicited in the telencephalon during the experiments recruiting voluntary attention only would be greater than those in the mesencephalon, because of top-down control in voluntary attention; (2) on the contrary, if reflexive attention exists, the N1 amplitudes evoked in the mesencephalon during the experiments recruiting reflexive attention only would be greater than those in the telencephalon because of bottom-up control in reflexive attention; and (3) if auditory perception depends on both voluntary attention and reflexive attention, the predictable stimuli would evoke both SPN and N1.

## 2. Materials and Methods

### 2.1. Animals

Sixteen frogs (8 females and 8 males) were captured during their reproductive season in the Emei mountain area of Sichuan, China. The animals were separated by sex in two opaque plastic tanks (54 × 40 cm and 33 cm deep), which were placed in a room under a 12:12 light–dark cycle (light on at 08:00 am). The temperature and humidity in the room were controlled at 23 ± 1 °C and 79.3 ± 8.5%, respectively. At the time of surgery, the mean mass of the animals was 9.96 ± 1.64 g, while the mean length was 4.65 ± 0.24 cm. The animals were fed with fresh crickets every three days.

### 2.2. Surgery

All electrophysiological experiments were carried out during the reproductive season of this species. Surgical procedures were described in detail in previous studies [53,54]. Briefly, the animals were anesthetized with 0.15% solution of tricaine methanesulfonate (MS-222) before surgery, and seven stainless steel screws (0.8 mm in diameter) were implanted into each animal’s skull with about 0.8 mm depth of the tips resting on the dura mater (Figure 1). Six of them were located on both sides of the telencephalon, diencephalon and mesencephalon, respectively, and the reference electrode was located above the cerebellum because of its several-fold lower activities compared with the cerebral ones [55]. Each electrode lead was formvar-insulated nichrome wire. One end of the wire was tightly enwound around the screw, while the other was tin soldered to a female pin of an electrical connector (the male pin was connected to the cable of the signal acquisition system). The electrodes were fixed to each animal’s skull with dental acrylic. The connector covered with self-sealing membrane (Parafilm® M; Chicago, IL, USA) was located approximately 1 cm above the animal’s head [56]. The experiments were carried out after 7 days of recovery after surgery. After finishing the experiments, the frogs were euthanized with an overdose of anesthetic, and then hematoxylin dye was injected into the skull holes where the electrodes were implanted previously to determine whether the electrodes were implanted at the correct locations, in order to verify that the EEG recordings were acquired from appropriate brain regions.

### 2.3. Recording Conditions

An opaque experimental tank (80 × 60 cm and 55 cm deep) containing mud and water was placed in an electromagnetically shielded and soundproof chamber (background noise was 23.0 ± 1.7 dB). An infrared camera with motion detection was mounted approximately 1 m above the tank for monitoring movement status of the animals. A signal acquisition system (Chengyi, RM6280C; Chengdu, China) was used to record the subjects’ electrophysiological signals. The band-pass filter was set at 0.16–100 Hz, while the sampling frequency was set at 1000 Hz.

### 2.4. Stimulus and Procedure

Five stimuli were used in the present study: white noise, pure tone of 1000 Hz, conspecific male advertisement call, screech call and silence. Because the results of statistical analyses in animal behavior, neuroscience, and ecological studies might be affected by pseudoreplication [57,58,59,60], we used multiple stimulus exemplars to control these possible effects. Specifically, we randomly selected four conspecific advertisement calls that contained five notes and were recorded from four different individuals inside their burrows. The temporal and spectral parameters of the selected advertisement calls were close to the averages for the population. Since we encountered only one individual that was attacked by a snake in a field, only one screech call containing five “notes” was used in the present study. Both white noise and pure tone were constructed as a consecutive “call” with their duration equal to the average duration of the four advertisement calls (about 1.28 s) and shaped with a rise and fall time sinusoidal period of 7.5 ms (Figure 2).

White noise and one of the other types of acoustic stimuli but not silence was paired, i.e., white noise vs. target sound. For voluntary attention, each of the three types of stimulus pairs was selected randomly and presented antiphonally with 1.5 s inter-stimulus intervals (ISI). After 20 presentations to familiarize the subjects with the patterns of stimulus sequences, the target sound at the last position of every N presentation of this sound was replaced by silence until 120 replacements were achieved (N = 3–6, each number was selected 30 times but selected randomly for each replacement). Thus, for each stimulus pair and each frog, a total of 1320 sound presentations with white noise were presented 660 times, the target sounds were presented 540 times, and the replacements happened 120 times. The session lasted about 66 min and included 4 blocks with 5 min breaks between blocks in order that the animals could have a rest. To test reflexive attention, each of the three types of stimulus pairs was selected and presented randomly using an equiprobability paradigm, in which both white noise and the target sound were presented in 50% probability. The ISI was set randomly at one of 1.1, 1.3, 1.5, 1.7 and 1.9 s for each presentation. Therefore, the subjects could not predict what the next presentation was and when the next presentation would appear; thus, voluntary attention for a given stimulus would be eliminated to an extreme. For each stimulus pair and each animal, a total of 200 stimulus presentations with each stimulus played back 100 times were presented in a random order. The session lasted about 9 min. For the stimulus pairs including advertisement calls, each stimulus pair was broadcasted to four animals (two females and two males), and all animals had never heard the acoustic stimuli before.

Acoustic stimuli were played back using two portable speakers (SME-AFS; Saul Mineroff Electronics, Elmont, New York, NY, USA) that were placed equidistantly at the opposite ends of the experimental tank. The sound pressure was adjusted to 65 ± 0.5 dB SPL for each acoustic stimulus using a sound pressure meter (Aihua, AWA6291; Hangzhou, China; re 20 µPa, fast response, C-weighting), measured at the center of the experimental tank, approximately equal to the average of natural sound pressure level of male calls. Thus, the sound level distribution at the experimental bank bottom was close to a quasi-free sound field. Furthermore, the animals always remained motionless at one corner of the experimental tank throughout the experiments. Accordingly, it was highly unlikely that the ERP measures would be affected significantly by the tiny differences in the stimulus amplitude across the tank bottom. All experimental procedures were realized with a custom-made software written in C++, which could automatically save the order of the random stimulus stream. A trigger pulse was sent to the signal acquisition system at every stimulus onset via the parallel port of a PC for further time-locking analysis.

### 2.5. Data Acquisition and Processing

After recovery for 7 days, each animal was placed in the tank and connected to the signal acquisition system for habituation about 1 day before the following experiments. Then, the EEG signals and behavioral data were recorded according to the above described experimental paradigms. In order to extract ERP components, the EEG raw data were filtered using a band-pass filter of 0.25–25 Hz and a notch filter of 50 Hz. For the experiments testing voluntary attention, EEG signals were divided into epochs with duration of 700 ms, including a pre-stimulus baseline of 200 ms, for the target sound. To analyze the SPN component, EEG signals were divided into epochs with duration of 2980 ms, from 200 ms pre-presentation of white noise to the presentation of silence. In order to test whether auditory perception depended on both voluntary attention and reflexive attention, EEG signals were divided into epochs with duration of 3480 ms, from 200 ms pre-presentation of white noise to 500 ms after the onset of presentation of target sound. For the experiments testing reflexive attention, EEG signals were divided into epochs with duration of 700 ms, including a pre-stimulus baseline of 200 ms. All epochs were visually inspected, and those with artifacts in which the maximal amplitude exceeded ±60 μv were removed from further analysis. Accepted trials (roughly 55% for each stimulus pair and each brain region) were averaged according to stimulus type for each brain area within each session.

For each acoustic stimulus and each brain region, the peak of each ERP component could be found in the grand average waveforms that were acquired from averaged waveforms across all frogs (see Appendix A). For all experiments, the latency of the N1 peak was measured from the grand average waveforms for each brain area and each stimulus; then, the median was calculated regardless of brain area and acoustic stimulus. Finally, the time window of the N1 component was defined as the latency range of 20–120 ms after the target stimuli onset with the median as the midpoint. The N1 amplitude was calculated as the mean amplitude in that time window using a custom-made software in Matlab. Similarly, for the experiments recruiting voluntary attention, the SPN amplitude was defined as the mean amplitude during intervals of 500 ms before the onset of silence replacement. For each ERP component, the latency was calculated as the half area latency with the same time window as the amplitude measurement, i.e., computing the area under the ERP waveform over a given latency range (i.e., time window) and then finding the time point that divides that area into equal halves using a custom-made software in Matlab [30]. Because we focused on detecting the direction of information flow (top-down or bottom-up), the amplitudes or latencies of each ERP component were averaged over the left and right sides of the telencephalon, diencephalon and mesencephalon, respectively. Human-like auditory P2 and P3 components have been verified in the music frog [43,44]; however, these two components might link to other brain functions rather than attention. Accordingly, we did not consider these components in this study.

### 2.6. Statistical Analyses

Shapiro–Wilk *W* test and Levene’s test were used to estimate the distribution normality and homogeneity of variance for amplitudes and latencies of each ERP component. For the stimulus pairs including conspecific calls, latencies and amplitudes of ERP components were analyzed statistically using a three-way repeated measures ANOVA with the variables of “stimulus pair” (the four stimulus pairs including different conspecific calls), “sex” (female/male), and “brain area” (the telencephalon, diencephalon and mesencephalon). There was no significant main effect of “stimulus pair”, congruent with the idea that the four stimulus pairs were not significantly different at evoking responses from the animals. Thus, amplitudes or latencies of each ERP component for the stimulus pairs including conspecific calls were pooled regardless of “stimulus pair”. A three-way repeated measure analysis of ANOVA was used for the amplitudes and latencies of N1 and SPN components with the variables of “sex” (female/male), “acoustic stimulus” (conspecific call, pure tone and screech call), and “brain area” (the three brain areas). Both main effects and interactions for the variables were examined. Multiple comparisons using the Bonferroni correction and simple effect analysis were performed when ANOVAs returned a significant difference and the interaction effects were significant, respectively [61]. If the assumption of sphericity was violated, the Greenhouse–Geisser ε values were employed. The partial *η*^2^ value was used to determine the effect size (partial *η*^2^ = 0.20 was set as small, 0.50 as medium, and 0.80 as large effect size, respectively) [62]. SPSS software (release 21.0) was utilized for the statistical analysis using *p* < 0.05 as the significance level.

## 3. Results

### 3.1. The Results of the Experiments Recruiting Voluntary Attention

For SPN amplitudes (superimposition according to the onset of white noise before silence replacement), the main effect for the factor “brain area” rather than other factors was significant (*F**_2_*_,__28_ = 6.281, *ε* = 0.928, partial *η*^2^ = 0.164, *p* = 0.006; Table 1). The SPN amplitudes evoked in the telencephalon were significantly greater than those in the mesencephalon (Table 1). The interaction between “acoustic stimulus” and “brain area” was significant (*F*_4,__56_ = 2.582, *ε* = 0.666, partial *η*^2^ = 0.102, *p*
*=* 0.047; Table 1). Simple effect analysis showed that the SPN amplitudes in the telencephalon and diencephalon evoked by conspecific calls were significantly greater than those in the mesencephalon (Table 2; Figure 3a). The SPN latency did not show any significant main effect for all factors (Table 1).

For N1 amplitudes (superimposition according to the onset of the target stimuli), the main effect for the factor “brain area” rather than other factors was significant (*F**_2_*_,__28_ = 20.859, *ε* = 0.596, partial *η*^2^ = 0.598, *p*
*<* 0.001; Table 1). The interaction between “acoustic stimulus” and “brain area” was significant (*F*_4,__56_ = 12.265, *ε* = 0.375, partial *η*^2^ = 0.467, *p*
*=* 0.001; Table 1). Generally, the N1 amplitudes evoked in the mesencephalon were greater than those evoked in the telencephalon and diencephalon regardless of acoustic stimuli (Table 2; Figure 3b). For the N1 latencies, there was no significant main effect for all factors (Table 1).

### 3.2. The Results of the Experiments Recruiting Reflexive Attention

When the playback sequence of acoustic stimuli could not be predicted, statistical analysis for the N1 amplitudes found that the main effect for the factor “brain area” rather than other factors was significant (*F**_2_*_,__28_ = 14.053, *ε* = 0.691, partial *η*^2^ = 0.501, *p* = 0.001; Table 3). In addition, the interaction between “acoustic stimulus” and “brain area” was significant (*F*_4,__56_ = 10.263, *ε* = 0.49, partial *η*^2^ = 0.423, *p*
*=* 0.001; Table 3). The N1 amplitudes in the telencephalon elicited by screech call were significantly greater than those by the conspecific calls (Table 4; Figure 4). Generally, the amplitudes of N1 elicited in the mesencephalon were significantly greater than those in other brain areas regardless of acoustic stimuli (Table 4; Figure 4). The N1 latency did not show any significant main effect for all factors (Table 3).

## 4. Discussion

### 4.1. Voluntary Attention Involved in Auditory Perception of Music Frogs

Voluntary attention is crucial to survival for many creatures, because voluntary attention can enhance perceptual representation and make attention resources more concentrated [63,64]. For example, voluntary attention in predators can increase the probability of predation by making them more focused on their target prey [1]. Voluntary control of cognition, such as voluntary attention, operates based on anticipatory information [4] and is controlled by top-down signals within the brain [35,36]. In humans, the subjects’ expectation and prediction of an oncoming stimulus, related to voluntary attention, could be reflected by the SPN component [35,65]. Consistent with this, the current results show that the silence period replacing the predictable acoustic stimuli could evoke greater SPN in the forebrain, especially in the telencephalon. Forebrain networks control all forms of attention based on task demands and the physical salience of stimuli [1]. Moreover, these brain networks contain neural circuits that distribute top-down signals to sensory processing areas and enhance information processing in those areas. Due to important neuroanatomical features including a set of brain structures that attention depends on that have been conserved during vertebrate brain evolution [48,49,50], similar SPN components across different species may indicate similar brain functions to some extent. Consequently, it seems reasonable to speculate that music frogs have the ability of voluntary attention. This speculation has been confirmed by previous studies showing that male music frogs consistently avoid producing advertisement calls that would overlap other sounds and generally produce calls in advance of rivals’ conspecific calls [52], and female music frogs prefer inside calls to outside ones in the phonotaxis experiments [51].

For most anuran species, various species always gather in choruses to attract conspecific mates, during which the high levels of noise background may interfere with the transmission and reception of sound signals, thus affecting communication efficiency [18,20]. For example, overlapping calls may obscure fine acoustic attributes of the calls and further affect signal selectivity and decision-making in females [19,20]. In order to mitigate these negative effects, some male anurans use selective attention to adjust note/call timing with respect to only one or two neighbors with the loudest (or nearest) calls [24,25] or with calls containing more biological significance in the chorus [52]. By the “inhibitory-resetting” mechanism of call-timing, a male can increase the likelihood of occupying a leading position relative to those of his neighbors, with which the male may gain a mating advantage due to the precedence effect [28,29]. In fact, computer modeling has demonstrated that the “inhibitory-resetting” mechanism and selective attention may be favored by selection when female choice is biased by the precedence effect [27]. Consequently, voluntary attention may play a very important role in reproductive success in anurans.

### 4.2. Reflexive Attention Also Involved in Auditory Perception of Music Frogs

Reflexive attention is triggered rapidly by the perceived particular stimuli or stimulus attributes and is sensitive to the physical properties of the stimulus [4,63]. Different from voluntary attention, reflexive attention is driven by bottom-up control [4,66]; however, it is also widespread in various taxa [1]. For example, field playback studies have shown that playbacks of low-frequency calls that resemble those produced by large males to male natterjack toads (*Bufo calamita*) induces males to move away from the speaker [67], and male frogs in some species may increase, reduce or stop calling when exposed to the rivals’ calls [68,69,70]. Birds that are searching for food will quickly stop searching and glance at their surroundings when an alarm call is suddenly played back to them [12]. These behavioral responses are assuredly derived from perception of unexpected sounds, including reflexive attention.

The auditory N1 component, sensitive to the physical properties of stimulus, is correlated with attention [33,34]. Specifically, compared with unattended sounds, attended sounds can elicit greater N1 amplitudes, suggesting that the N1 amplitude can reflect auditory attention by which the brain can selectively attenuate further processing for the unattended sounds [71]. The current results show that N1 amplitudes elicited in the mesencephalon were significantly higher than those in both the telencephalon and diencephalon, and that there were significant differences in N1 amplitudes in the telencephalon evoked by different stimuli. Because the sequence of acoustic stimuli could not be predicted in the experiments recruiting reflexive attention only, the N1 amplitudes with the mesencephalic dominance indicated that the music frogs had the ability of reflexive attention. Consistent with this, our field work found sudden sounds, such as white noise, barks, and birdcalls as well as conversation could result in the frogs suddenly decreasing or stopping calling. The present results are also in agreement with previous electrophysiological studies that have demonstrated that N1 amplitudes are modulated by the biological significance of the acoustic stimuli [43,44,72]. Although some brain regions, including the supratemporal plane, lateral aspect of the temporal and parietal cortex and the motor and premotor cortices, are proposed as the main origins of N1 components in humans [34,73], the origins of N1 components in the frogs remain to be further elucidated.

### 4.3. The Combination of Voluntary Attention and Reflexive Attention May Guide Auditory Perception in Music Frogs

The present results show that both SPN and N1 components could be evoked when the sequence of acoustic stimuli could be predicted (Appendix A), suggesting that the auditory perception of music frogs might simultaneously invoke both voluntary attention and reflexive attention. Based on the former, males could compete effectively with other rivals and females could choose mates more accurately. On the other hand, reflexive attention allows the animals to respond quickly to unexpected sounds and/or unexpected urgent situations, which is crucial for survival. Under natural conditions, animals need to watch out for predators while they focus on courtship or foraging. For example, the Eastern rat snakes (*Elaphe obsoleta*) like to raid Golden-cheeked warbler (*Dendroica chrysoparia*) nests while the adult birds are away for foraging, so the slightest sound or movement around the nest will cause the adults to return [74]. Consistent with this, the amplitudes of N1 evoked in the music frogs’ telencephalon by the screech call (associating with survival) were significantly greater than those evoked by the conspecific calls. Thereby, both sexes can be alert to dangerous signals in the environment and avoid the risk of predation, which is beneficial for survival and reproduction.

The push–pull fashion of interaction between voluntary attention and reflexive attention in the struggle to control the focus of attention resources has been demonstrated in higher vertebrate taxa [2,8,9,10,11]. Similarly, we found that the auditory perception of the music frogs might invoke both voluntary attention and reflexive attention. These findings, including the present results, indicate a phylogenetically early emergence of interaction between voluntary attention and reflexive attention in animals. Since the beginning of vertebrate brain evolution, neural mechanisms of attention have selected the information that can gain access to the brain networks that make cognitive decisions [1], during which information selection dominated by the forebrain networks depends on the goals and tasks of the animal while information selection dominated by the midbrain networks depends on stimulus salience for the animal [50,75,76]. The relative dominance of the forebrain and midbrain networks may change across species: there is an advantage for the forebrain networks to dominate information selection in the context of hunting or mate searching; conversely, there is an advantage for the midbrain network to dominate in monitoring the environment for unexpected stimuli in order to avoid predation [1]. Although the brain structures (especially the forebrain) of lower vertebrates are less differentiated than those of higher mammals [77], important neuroanatomical features and brain functions have been preserved during the evolution of vertebrate brains [50,78]. For example, the frontal cortex may play an important role in auditory attention modulation in humans, non-human primates and rodents [79,80,81], while birds also have similar structures involved in attention regulation [82]. Correspondingly, our previous study found that the telencephalon might play a similar role in attention regulation in music frogs as the frontal cortex did in mammals [83]. Therefore, it seems reasonable to speculate that similar auditory attention mechanisms (voluntary attention vs. reflexive attention) may be widespread in vertebrates.

## 5. Conclusions

In summary, we found that the greatest SPN amplitudes could be evoked in the telencephalon when the sequence of acoustic stimuli could be predicted, while the greatest N1 amplitudes could be evoked in the mesencephalon when the sequence of acoustic stimuli could not be predicted. These findings suggest that both voluntary attention and reflexive attention may exist and engage in auditory perception in music frogs. However, we could not exclude two main limitations of this study. Firstly, we did not measure auditory attention directly using behavioral paradigms. Secondly, we discussed the biological significance of SPN/N1 based on corresponding findings in humans. Accordingly, behavioral responses to various acoustic signals should be quantified in future research in order to determine their relevance to the electrophysiological responses found here. Moreover, future studies are needed for a comprehensive understanding of similarities in brain functions (between mammals and frogs) and to what extent they are reflected by each ERP component at both the behavioral and electrophysiological levels.

## Figures and Tables

**Figure 1 biology-11-00879-f001:**
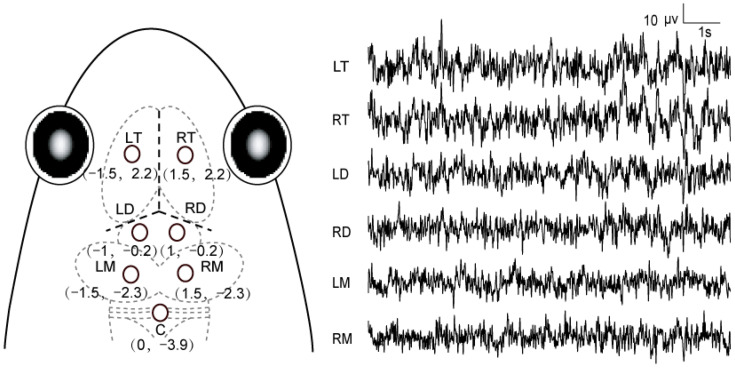
Schematic diagram showing electrode placements and 10 s typical raw unfiltered EEG tracings for each channel. Three dashed lines in bold indicate the intersection among the three suture lines in the frog skull. The first value for each electrode location was the mediolateral distance between this site and the midline, while the second one was the anteroposterior distance between the site and the intersection. LT and RT, both sides of the telencephalon; LD and RD, both sides of the diencephalon; LM and RM, both sides of the mesencephalon; C, the cerebellum.

**Figure 2 biology-11-00879-f002:**
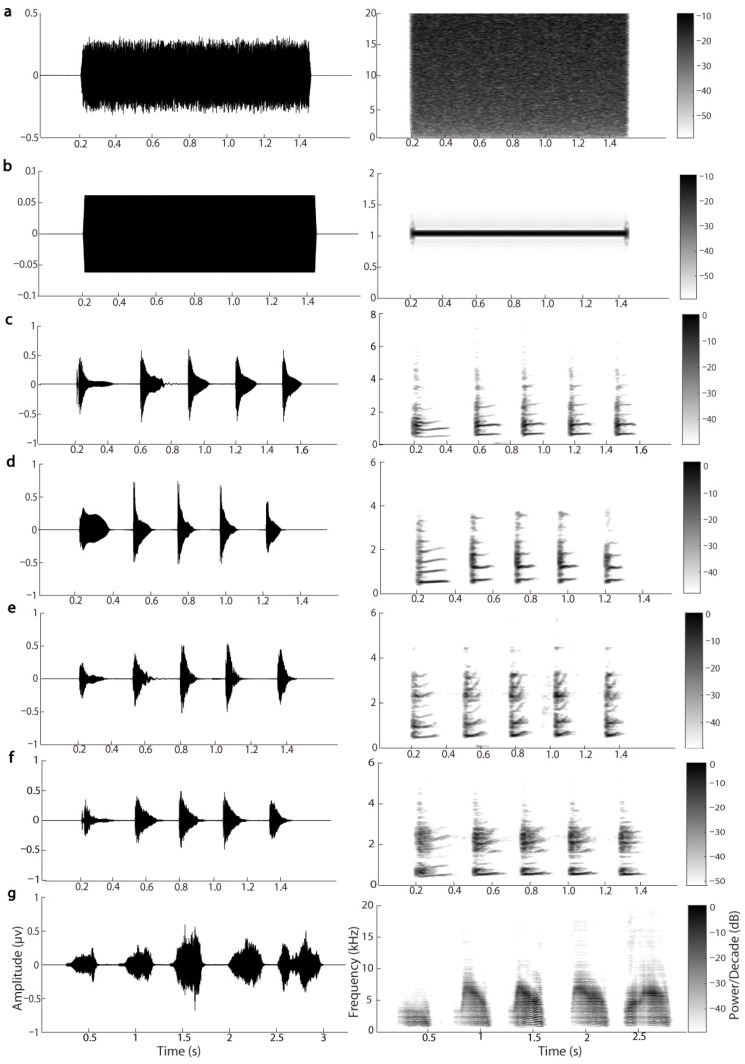
Waveforms and spectrograms of the seven acoustic stimuli: (**a**) white noise, (**b**) 1000 Hz pure tone, (**c**–**f**) four conspecific advertisement calls from different individuals, (**g**) screech call.

**Figure 3 biology-11-00879-f003:**
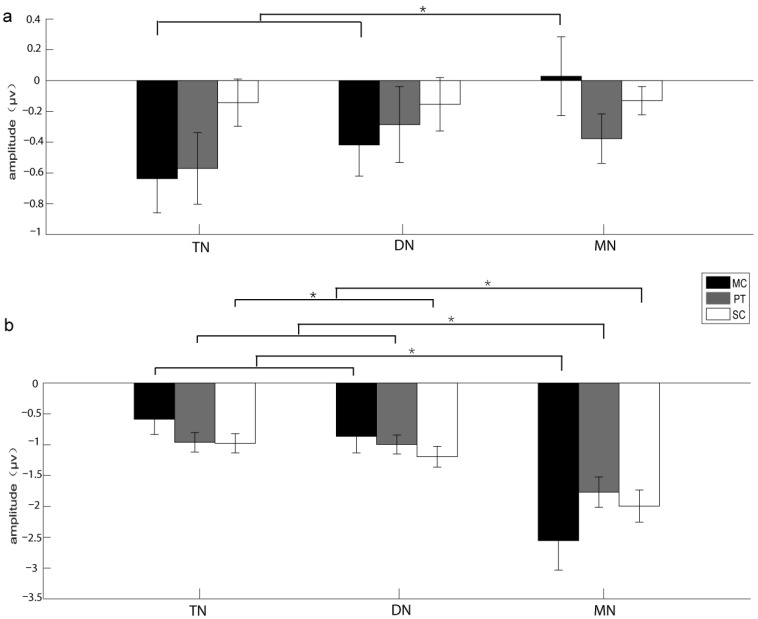
The amplitudes and standard errors of SPN (**a**) and N1 (**b**) components for the experiments recruiting voluntary attention. Each asterisk (*) indicates significant differences among different brain areas (*p* < 0.05). Note that SPN was acquired from superimposition of the recorded waveforms according to the onset of white noise before silence replacement, while N1 was acquired from superimposition of the recorded waveforms with respect to the onset of the target stimuli. MC, conspecific male advertisement call; PT, 1000 Hz pure tone; SC, screech call; TN, telencephalon; DN, diencephalon; MN, mesencephalon.

**Figure 4 biology-11-00879-f004:**
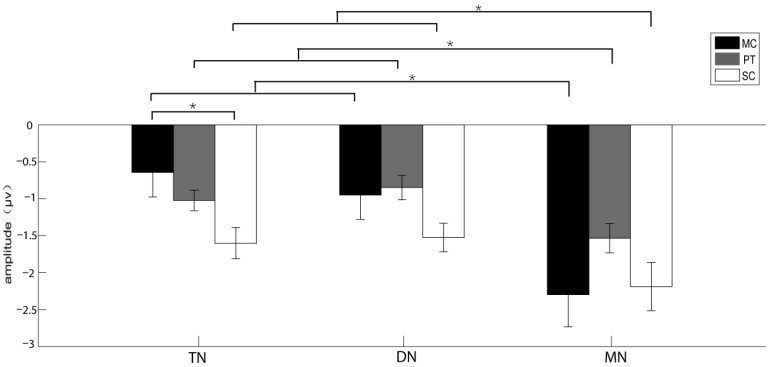
The amplitudes and standard errors of N1 components for the experiments recruiting reflexive attention. Each asterisk (*) indicates significant differences between different acoustic stimuli and different brain areas (*p* < 0.05). MC, conspecific male advertisement call; PT, 1000 Hz pure tone; SC, screech call; TN, telencephalon; DN, diencephalon; MN, mesencephalon.

**Table 1 biology-11-00879-t001:** Results of ANOVA for the amplitudes and latencies of SPN and N1 with respect to the three factors for the experiments recruiting voluntary attention.

	For the Amplitude (1,14)(2,28)(2,28)(4,56)					For the Latency (1,14)(2,28)(2,28)(4,56)				
	*F*	*ε*	*p*	Partial *η*^2^	MCBC	*F*	*ε*	*p*	Partial *η*^2^	MCBC
SPN										
Sex	1.994	NA	0.18	0.125	NA	0.177	NA	0.681	0.012	NA
Acoustic stimulus	0.783	0.905	0.467	0.053	NA	2.391	0.925	0.110	0.146	NA
Brain area	6.281	0.928	0.006 *	0.164	TN > MN	0.893	0.845	0.421	0.06	NA
Acoustic stimulus*Brain area	2.582	0.666	0.047 *	0.102	see Table 2	1.343	0.688	0.266	0.088	NA
N1										
Sex	0.677	NA	0.424	0.046	NA	0.021	NA	0.886	0.005	NA
Acoustic stimulus	0.235	0.878	0.792	0.017	NA	3.538	0.65	0.067	0.202	NA
Brain area	20.859	0.596	0.000 **	0.598	MN > TN, DN	1.357	0.711	0.272	0.088	NA
Acoustic stimulus*Brain area	12.265	0.375	0.001 *	0.467	see Table 2	3.627	0.409	0.051	0.206	NA

Note: The degrees of freedom corresponding to the three factors and interaction are shown in the first brackets. The symbol “>” means the amplitudes on the left side of “>” are significantly greater than those on the right side, and no significant difference exists among the corresponding conditions on the same side of “>”. Note that SPN was acquired from superimposition of the recorded waveforms according to the onset of white noise before silence replacement, while N1 was acquired from superimposition of the recorded waveforms according to the onset of the target stimuli. *F*, the *F* value from ANOVA; partial *η*^2^, effect size for ANOVA; *ε*, values of epsilon of Greenhouse–Geisser correction; MCBC, multiple comparisons with the Bonferroni correction; NA, not applicable; TN, telencephalon; DN, diencephalon; MN, mesencephalon. *, *p* < 0.05 and **, *p* < 0.001.

**Table 2 biology-11-00879-t002:** Results of simple effect analysis for the amplitudes of N1 for the experiments recruiting voluntary attention.

	*F*	*p*	Partial *η^2^*	MCBC
SPN				
Brain area|MC	6.331	0.012 *	0.493	TN, DN > MN
N1				
Brain area|MC	11.236	0.001 *	0.634	MN > TN, DN
Brain area|PT	9.543	0.003 *	0.595	MN > TN, DN
Brain area|SC	8.754	0.004 *	0.574	MN > DN > TN

Note: The symbol “|” denotes the conditions on the left side of “|” under the conditions on the right side of “|”. The symbol “>” means the amplitudes on the left side of “>” are significantly greater than those on the right side, while no significant difference exists among the corresponding conditions on the same side of “>”. *F*, the *F* value from ANOVA; partial *η*^2^, effect size for ANOVA; MCBC, multiple comparisons with the Bonferroni correction; MC, conspecific male advertisement call; PT, 1000 Hz pure tone; SC, screech call; TN, telencephalon; DN, diencephalon; MN, mesencephalon. *, *p* < 0.05.

**Table 3 biology-11-00879-t003:** Results of ANOVA for amplitudes and latencies of N1 with respect to the three factors for the experiments recruiting reflexive attention.

	For the Amplitude (1,14)(2,28)(2,28)(4,56)					For the Latency (1,14)(2,28)(2,28)(4,56)				
	*F*	*ε*	*p*	Partial *η*^2^	MCBC	*F*	*ε*	*p*	Partial *η*^2^	MCBC
N1										
Sex	0.689	NA	0.42	0.047	NA	0.735	NA	0.406	0.05	NA
Acoustic stimulus	3.183	0.97	0.057	0.185	NA	2.526	0.782	0.098	0.153	NA
Brain area	14.053	0.691	0.001 *	0.501	MN > TN, DN	0.843	0.645	0.485	0.057	NA
Acoustic stimulus*Brain area	10.263	0.49	0.001 *	0.423	see Table 4	0.602	0.446	0.68	0.041	NA

Note: The degrees of freedom corresponding to the three factors and interaction are shown in the first brackets. The symbol “>” means the amplitude on the left side of “>” is significantly greater than those on the right side, and no significant difference exists among the corresponding conditions on the same side of “>”. *F*, the *F* value from ANOVA; partial *η*^2^, effect size for ANOVA; *ε*, values of epsilon of Greenhouse–Geisser correction; MCBC, multiple comparisons with the Bonferroni correction; TN, telencephalon; DN, diencephalon; MN, mesencephalon. *, *p* < 0.05.

**Table 4 biology-11-00879-t004:** Results of simple effect analysis of the amplitudes for N1 for the experiments recruiting reflexive attention.

	*F*	*p*	Partial *η^2^*	MCBC
Stimulus|TN	4.148	0.040 *	0.39	SC > MC
Brain area|MC	10.147	0.002 *	0.391	MN > TN, DN
Brain area|PT	5.502	0.019 *	0.771	MN > TN, DN
Brain area|SC	5.823	0.016 *	0.688	MN > TN, DN

Note: The symbol “|” denote the conditions on the left side of “|” under the conditions on the right side of “|”. The symbol “>” means the amplitudes on the left side of “>” are significantly greater than those on the right side, and no significant difference exists among the corresponding conditions on the same side of “>”. *F*, the *F* value from ANOVA; partial *η*^2^, effect size for ANOVA; MCBC, multiple comparisons with the Bonferroni correction; MC, conspecific male advertisement call; PT, 1000 Hz pure tone; SC, screech call; TN, telencephalon; DN, diencephalon; MN, mesencephalon. *, *p* < 0.05.

## Data Availability

The dataset generated and analyzed in the current study is available at http://dx.doi.org/10.17632/3r9rdw5mfx.1 (accessed on 20 April 2022).

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
