# Peer review of "Possible Event-Related Potential Correlates of Voluntary Attention and Reflexive Attention in the Emei Music Frog"

_biology, 2022, doi:10.3390/biology11060879_

Round 1

Reviewer 1 Report

This paper describes experiments on  "voluntary attention" and "reflexive attention" in music frogs.  The main flaw in the authors' conceptualization of their work is that they are not actually measuring attention (as they imply in the title by use of the phrase "guiding auditory perception").  At most, they can say that they have quantified certain ERP/SPN correlates or signatures of what may be attentional processes.  Frogs in their study did not exhibit any differential behavioral responses to the presented stimuli, as would be expected if they were paying "attention."  In fact, the authors state that the frogs "usually remained motionless" (line 219) during the experiments.   Thus, interpretation of the data in terms of attention is not warranted -- it cannot be concluded that their data meet their goal of determining how 2 forms of attention "interact dynamically during auditory perception" (line 65), because perception was not quantified.

Specific comments:

1. line 96: Those peak latencies are derived from human data.  ERP components recorded from animals can have different latencies.  There are considerable data on ERPs in animals, including frogs, that are not cited that would demonstrate these species differences in component latencies.  

2. Line 156: "eliminate unexpected data." What does this mean?

3. Fig. 1: Please give the scale of the head.  Are these tracings raw unfiltered EEG?

4. Why was the bandpass filter set to 0.05-200 Hz? (line 168).

5. Fig. 2: are these waveforms and spectrograms from the electronic stimuli? The authors should show instead the acoustic stimuli at the frogs' ears.

6. Line 201: What is meant by fatigue and how was it quantified?

7. Explain how the screws on the skull were connected to the signal acquisition system (line 228).

8. Data are presented in the form of tables.  But, actual filtered ERP and SPN components are not shown.  This is a major flaw.  Without seeing these waveforms, it is not possible to evaluate the scientific soundness of the data.

9. Why was the N1 component defined within the large latency range of 20-120 ms?  This is not consistent with other ERP work. 

10.  The authors state that "it seems reasonable to speculate that the music frogs have the ability of voluntary attention."  (line 396).  One can indeed make this speculation, but it is not supported by the presented data. 

Reviewer 2 Report

Specific comments

fig 1-  indicate -2.2

line 78 is better or not and. It has been shown that a frog can choose a more distant partner even if at the point of perception its signal was weaker than the signal of the nearest male.

Lines 93-102 - it is necessary to indicate that this material was obtained on mammals and, in my opinion, the text is too detailed. N1 is absolutely different in frogs and mammals

 Lines 148-155 - I do not quite understand where the recording electrodes were located in the bone or on the surface of the brain, or whether the fool was pierced. The position of the reference electrode was located above the cerebellum also raises doubts; it seems that such an electrode can record the activity of the lateral lemniscus and the dorsal nucleus of the medulla oblongata.

Line 231- It looks like that the low-pass at 25Hz filter too low?

Line 240 “All EEG epochs were inspected visually and those with muscle artifacts and electrode drifts were removed from further analysis” How many  events (in %) have been removed?

Lines 247-248   “For the experiments recruiting voluntary attention, the SPN component was defined as 247 the mean amplitude during intervals of 500 ms before the onset of silence replacement”. It’s looked more as a power or energy but not as amplitude.

Lines 250-254 it seems to me impossible to discuss the origin of the recorded potential peaks by referring to the data obtained on humans.

Lines 272=276 I recommend giving references to the specific statistical methods used

Lines 432-434 So where does N1 appear, or does it occur simultaneously in all nuclei?

 Fig 3a  I don't see a statistical difference between LD and RM even though it's shown in the picture. For what from these three signals these differences were  calculated? And where are Y coordinates? Is it true that all EP were less than 1 mkV?

Lines440-442 "The present results are also consistent with previous elec-440 trophysiological studies that have shown that the conspecific calls always elicit greater 441 N1 amplitudes compared with other non-conspecific sounds [40, 60, 61]." Unfortunately I could'see this results evidenly.

Fig 4 the capture presented twice. The abbreviations ( the same as in fig 3) could be omitted. The last note should be applicable for all tables.

My general suggestions: 1. It is desirable to indicate where in relation to the brain (inside the brain, in the bone, in the perilymph) the recording electrodes were located. It allows to judge about their spatial selectivity. It is quite possible that in all cases the same activity was recorded, and the amplitude differences were determined only by the mutual impedance of the electrodes. In any case, the location of the indifferent electrode in the region between the medulla oblongata and the midbrain should be somehow justified, since just in this zone that the most powerful evoked response to any sounds are usually recorded. 2. It is necessary to  include in the main text the real illustrations of evoked potentials with the standard deviation, 3. In all cases, it is completely unacceptable to assume the same origin and the same properties of the potential N1 in the frog and in mammals. 4. Simplify the number of statistical comparisons, limited them to standard ones. 5 Discuss all described statistically significant differences, including between the right and left sides of the same structure.

Reviewer 3 Report

In the article by Niu et al, an acoustic evoked potential approach was used to investigate different forms of attention in the music frog, an anuran that uses vocalizations for courtship and social interactions. Recordings of different sound stimuli were used in animals that were implanted with EEG electrodes to monitor sensory responses in hind brain, mid brain and forebrain regions. The authors found statistically significant differences in SPN and N1 components of evoked potentials, which provides evidence that supports their conclusion that voluntary and reflexive types of attention are present in this amphibian species.

The introduction is clear and informative to understand the background information leading to the motivation of the study.  The materials section is well written and should provide information to other readers who may want to use a similar approach to replicate or extend the findings reported.

The presentation of results is clear, however some information is missing or misplaced. For example Figure 3 is either missing data for white noise, or the abbreviation for white noise WN was misplaced (Line 311).

The statement in line 310 is not clear. Do authors mean that statistical comparisons are not shown for clarity of presentation?

The "Not Applicable=NA" abbreviation is misplaced in Tables 2 and 4.

The Discussion is well written and authors are careful to over interpret the data. The discussion cites relevant work that supports their careful considerations.

Round 2

Reviewer 1 Report

The authors' response to previous reviews consisted mainly of pointing out similarities in brain organization and function across vertebrate species, a point that is not in dispute and only tangenially relevant to the substance of the reviews.  The issue is not the existence of similarities across species, or whether there exist ERP correlates of attention in humans, but whether the authors can claim on the basis of this experiment that two types of attention "exist and engage in" (lines 12-13, 18-19) music frogs.  Because the frogs in this study did not make behavioral responses, it is not sound scientifically to interpret the data in terms of attentional processes.  At most, the authors can say that they identified possible ERP correlates of attentional-like processes.  For this reason, a better title would be "Possible ERP correlates of voluntary and reflexive attention in music frogs."  Some caveats to the results are acknowledged in the response to reviewers but in the manuscript.  Statements that go beyond the data persist; for example, lines 131-132; lines 451-452. I disagree that "these limitations should not affect the current conclusions severely" (line 527).  Argument by analogy with results from other species where behavior has been measured is not sufficient. The authors need to tone down their interpretation. 

I am still unclear how response amplitude and latency were measured.  The authors state only that they compiled grand average waveforms (across frogs? or across all stimulus repetitions within an individual frog?).   But how were amplitude and latency of the waveforms quantified -- manually? by an algorithm? Was a threshold criterion used to ensure that peak amplitude was above background noise?

I have concerns about the statistical tests. As I understand it, the authors performed 4 repeated measures ANOVAs -- N1 amp, N1 latency, SPN amp, SPN latency (line 287; should read analysis of variance, not analysis of ANOVA), followed by a series of post-hoc LSD tests.  But it appears that p values were not corrected for these multiple tests (line 297). Some of the significant p values would become non-significant if this appropriate statistical correction was used.  

Reviewer 2 Report

I agree that the authors have considered my and other reviewers comments quite seriously and have modified the manuscript accordingly. In my opinion, the article can be published in its present form.

Author Response

Thank you very much for your comments.

Round 3

Reviewer 1 Report

The authors have addressed most of the comments in my previous reviews.  Aside from some editing changes for English sentence structure, I am satisfied with the authors' revisions.